# Dichloromethylation of enones by carbon nitride photocatalysis

Stefano Mazzanti [1,3], Bogdan Kurpil[1,3], Bartholomäus Pieber [2], Markus Antonietti [1] & Aleksandr Savateev [1✉]

Small organic radicals are ubiquitous intermediates in photocatalysis and are used in organic synthesis to install functional groups and to tune electronic properties and pharmacokinetic parameters of the final molecule. Development of new methods to generate small organic radicals with added functionality can further extend the utility of photocatalysis for synthetic needs. Herein, we present a method to generate dichloromethyl radicals from chloroform using a heterogeneous potassium poly(heptazine imide) (K-PHI) photocatalyst under visible light irradiation for C1-extension of the enone backbone. The method is applied on 15 enones, with γ,γ-dichloroketones yields of 18–89%. Due to negative zeta-potential (−40 mV) and small particle size (100 nm) K-PHI suspension is used in quasi-homogeneous flow-photoreactor increasing the productivity by 19 times compared to the batch approach. The resulting γ,γ-dichloroketones, are used as bifunctional building blocks to access value-added organic compounds such as substituted furans and pyrroles.

[1] Department of Colloid Chemistry, Max Planck Institute of Colloids and Interfaces, Am Mühlenberg 1, 14476 Potsdam, Germany. [2] Department of Biomolecular Systems, Max Planck Institute of Colloids and Interfaces, Am Mühlenberg 1, 14476 Potsdam, Germany. [3]These authors contributed equally: Stefano Mazzanti, Bogdan Kurpil. ✉email: oleksandr.savatieiev@mpikg.mpg.de

**C**arbon nitrides (CNs) are "all-in-one" photocatalysts that mediate dozens of different photocatalytic reactions and enable bifunctionalization of (hetero)arenes in one pot[1]. The organic semiconductors have also been efficiently employed in a continuous flow system for chemical synthesis eliminating the last obstacle (poor light penetration in heterogeneous solid-liquid mixture) on the way to widespread applications in organic synthesis[2]. Because of their low cost, ease of synthesis and stability against reactive intermediates and photobleaching, CNs already play an important role as heterogeneous photocatalysts for organic transformations[3–5]. CNs are also very versatile, and can be tailored depending on the application by bandgap engineering at the atomic and molecular level[6,7].

Most photocatalytic reactions are based on single electron transfer between the reagents and the photocatalyst[8]. Therefore, reactive open shell species are ubiquitous intermediates in photocatalytic processes[9–11]. Small organic radicals, such as $CH_3$, $CF_3$, $CHF_2^1$, and perfluoroalky[12], $CH_3O^{13}$ etc. are used for the functionalization of the organic molecules in order to tune steric and electronic properties. Furthermore, the lipophilicity and metabolic stability of pharmaceuticals may be adjusted in this way[14,15]. Despite their importance for medicinal chemistry, $CF_3$, alkyl, and $CH_3O$ groups are chemically stable. Therefore, further diversification of the molecule at these newly formed sites is problematic. For example, cleavage of C–F bond in $CF_3$-group is extremely demanding[16]. The same applies to C–O in the $CH_3O$-group[17,18].

Conversely $CHCl_2$ radical from the pool of small organic radicals is synthetically more useful. It enables the installation of an electrophilic carbon, and the C–Cl bonds can be conveniently cleaved using weak nucleophiles. In other words, the $CHCl_2$ radical allows for $C_1$-extension of the substrate framework, while simultaneously adding a chemically active functionality[19]. From this point of view, the $CHCl_2$ radical can be regarded as a "small functional radical".

Despite the obvious synthetic utility of the dichloromethyl radical, literature is still lacking reactions using dichloromethyl moieties in conjugate additions—the kind of reaction resembling a traditional polar Michael addition. The latter was well studied in photoredox catalysis[20–23]. An example shown in Fig. 1a employs methyl groups in tertiary amines and C=C double bond as coupling partners. The chemistry of dichloromethyl radicals is restricted to a few examples, while such radicals are generated predominantly by catalyst containing rare precious metals or dangerous chemicals (Fig. 1b, c). Our alternative approach uses cheap heterogenous carbon nitride (CN) photocatalysts (1–10 Euro per gram on a gram-scale synthesis)[24] and have a low toxicity[25].

We hypothesized that chloroform can be used as atom efficient source of $CHCl_2$ radicals. Although chloroform readily gives dichlorocarbene in the presence of strong bases, we concede that photocatalyst will alter the path of chloroform decomposition. Formation of the dichloromethyl radical thereby may be achieved by one-electron reduction of chloroform followed by elimination of a chloride anion.

In order to trigger this process, we chose potassium poly (heptazine imide) (K-PHI), a member of the CN family[26]. Upon irradiation with visible light, metastable long-lived radicals are generated that have been used as a pool of electrons to reduce different substrates[27]. Earlier, we developed photocatalytic methods to synthesize thioamides[28], dibenzyl sulfanes[29], 1,3,4-oxadiazoles[3], N-fused pyrroles[30], cyclopentanes[27], and halogenated aromatic hydrocarbons using K-PHI[31]. In related works, the long-lived carbon nitride radicals were applied in the delayed evolution of hydrogen[32,33].

Due to the advantages of flow reactors[34,35], several types of such photoreactors employing carbon nitrides have been reported

**Fig. 1 Previous works related to the designed reaction are presented.**
**a** Conjugate addition to enones. **b** Cyclisation of trichloroacetamides. **c** Synthesis of dichlorinated oxindoles. **d** Photocatalytic reaction developed in this work.

—packed bed photoreactor[36], serial micro batch photoreactors[2], and triphasic flow photoreactor[37]. Due to relatively small particle size (average diameter 100 nm) and highly negative zeta-potential $(-40\,mV)^{38}$, K-PHI gives stable colloidal solution and has been used in quasi-homogeneous catalysis[39]. Due to these features colloidal solution of K-PHI can be used in simple plug-flow photoreactors that are designed for homogeneous reaction mixtures.

All in all, we present an unusual photocatalyzed radical addition of dichloromethyl radicals to enones to form a new C–C bond (Fig. 1d). In this approach chloroform is used as a source of dichloromethyl radicals. The reaction is catalyzed by K-PHI using blue light irradiation. Using the discovered reaction, we show that light scattering by semiconductor particulate strongly affects their performance in batch reactors limiting the scalability of such transformations. A nineteen times higher productivity is achieved using a dedicated flow photoreactor employing quasi-homogeneous K-PHI nanoparticles. Finally, dichloromethyl adducts, i.e., γ,γ-dichloroketones, are used to access bifunctional building blocks and several classes of heterocyclic compounds.

## Results

**Optimization of reaction conditions.** Along these arguments, we studied the designed reaction between chalcone **1a**, chloroform, tetrahydroisoquinoline (THIQ) as an electron donor and K-PHI as the photocatalyst (see SI for preparation and characterization of K-PHI, Supplementary Fig. 1). Dichloroketone **2a** was synthesized initially with 17% yield when 1 equivalent of THIQ was used (Table 1, entry 1). By increasing the amount of THIQ gradually to four equivalents, the yield of **2a** was increased to 51% (entry 3). However, even higher yield (62%) of **2a** was achieved by

**Table 1 Screening of reaction conditions.**

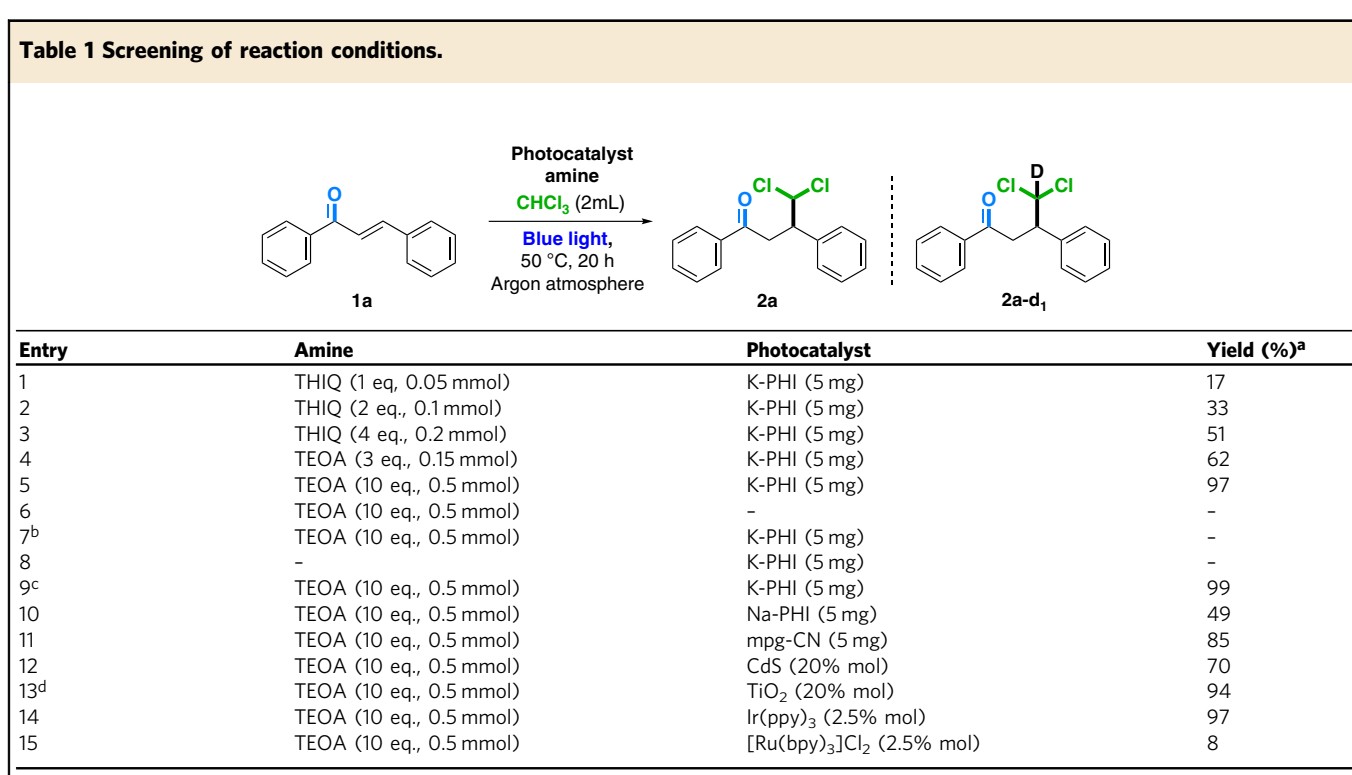

| Entry | Amine | Photocatalyst | Yield (%)[a] |
|---|---|---|---|
| 1 | THIQ (1 eq, 0.05 mmol) | K-PHI (5 mg) | 17 |
| 2 | THIQ (2 eq., 0.1 mmol) | K-PHI (5 mg) | 33 |
| 3 | THIQ (4 eq., 0.2 mmol) | K-PHI (5 mg) | 51 |
| 4 | TEOA (3 eq., 0.15 mmol) | K-PHI (5 mg) | 62 |
| 5 | TEOA (10 eq., 0.5 mmol) | K-PHI (5 mg) | 97 |
| 6 | TEOA (10 eq., 0.5 mmol) | – | – |
| 7[b] | TEOA (10 eq., 0.5 mmol) | K-PHI (5 mg) | – |
| 8 | – | K-PHI (5 mg) | – |
| 9[c] | TEOA (10 eq., 0.5 mmol) | K-PHI (5 mg) | 99 |
| 10 | TEOA (10 eq., 0.5 mmol) | Na-PHI (5 mg) | 49 |
| 11 | TEOA (10 eq., 0.5 mmol) | mpg-CN (5 mg) | 85 |
| 12 | TEOA (10 eq., 0.5 mmol) | CdS (20% mol) | 70 |
| 13[d] | TEOA (10 eq., 0.5 mmol) | $TiO_2$ (20% mol) | 94 |
| 14 | TEOA (10 eq., 0.5 mmol) | $Ir(ppy)_3$ (2.5% mol) | 97 |
| 15 | TEOA (10 eq., 0.5 mmol) | $[Ru(bpy)_3]Cl_2$ (2.5% mol) | 8 |

Reaction conditions: 1 eq., 0.05 mmol, 10.4 mg of **1a**; under light irradiation ($\lambda = 461$ nm, $51 \pm 0.03$ mW cm$^{-2}$), blue LED.
[a]Yields estimated by GC-MS.
[b]No light.
[c]Reaction performed in CDCl$_3$.
[d]Reaction performed under UV light ($\lambda = 365$ nm, $17.5 \pm 0.03$ mW cm$^{-2}$).

using 3 equivalents of triethanolamine (TEOA) as electron donor (entry 4). The optimum conditions include ten equivalents of TEOA, under which we achieved 97% yield (entry 5). The reaction does not proceed without catalyst, light or a sacrificial electron donor (entry 6–8). CDCl$_3$ is a suitable source of CDCl$_2$ radicals offering a route for $d$-labeled dichloroketones **2a-$d_1$** with 99% yield (entry 9). We also evaluated the robustness of the reaction using different alcohols as hole scavengers, observing the formation of the desired product in all cases, albeit in lower yield (Table S1, entry 11–14). These results illustrate the better ability of amines to donate electrons compared to alcohols, due to lower oxidation potentials (e.g., $+0.5$ V for TEOA, $+1.5$ V for benzyl alcohol and $+1.9$ V for MeOH, EtOH, $^i$PrOH (Supplementary Note 1). It is also supported by higher H$_2$ production rate over carbon nitride materials using TEOA as electron donor compared to MeOH and EtOH[40,41] and comparative tests of benzyl alcohol oxidation versus benzylamine[37,42]. Moderate heating (50 °C) facilitates the reaction, as the yield of **2a** was 64% when reaction was performed at 20 °C (Supplementary Table 1, entry 21). We also compared the catalytic activity of other materials and photoredox complexes. Na-PHI gave **2a** with 49% yield (entry 10)[43]. Similar behavior was already observed during the photocatalytic synthesis of thioamides[28]. Mesoporous graphitic carbon nitride (mpg-CN) gave **2a** with comparable yield 85% (entry 11). The inorganic semiconductors CdS and TiO$_2$ gave **2a** in 70 and 94% yield, respectively (entries 12,13). Homogeneous Ir(ppy)$_3$ gave **2a** with 97% yield (entry 14), while [Ru(bpy)$_3$]Cl$_2$ only resulted in 8% of **2a** (entry 15). Furthermore, it was also shown that recycled K-PHI remains photocatalytically active for at least two further cycles (Supplementary Table 2).

**Reaction scope**. Using the optimized conditions fifteen dichloroketones have been isolated with 18–89% yield (Figs. 2a–o). The

characterization of products was conducted by NMR analysis. Dichloroketones **2** do not give stable molecular ions in the mass spectra (electron ionization). For example, the expected $m/z$ value for **2a** is 292. However, a signal with $m/z$ 221 was detected (Supplementary Fig. 2). The latter corresponds to 2,4-diphenylfuran. In general 2,4-diarylsubstituted furans are products of oxygen nucleophilic attack at CHCl$_2$-group followed by elimination of two molecules HCl under the conditions of GC-MS data acquisition. Below we employ the reactivity of CHCl$_2$ group in synthesis of pyrroles and furans. Nonetheless, elemental analysis of **2a** revealed chlorine content in excellent agreement with the calculated content confirming the proposed structure. We further proved the product structure, using deuterated chloroform as dichloromethyl source, observing the rise of the triplet in the $^{13}$C NMR spectrum in the $d$-labeled compound (**2a-$d_1$**).

Dichloromethylated chalcones bearing strong electron withdrawing groups, i.e., CN–, MeO$_2$C–, and pyridin-2-yl, **1p–r**, gave the corresponding products **2p-r** in low yields as analyzed by $^1$H-NMR spectrometry of the crude reaction mixture (Supplementary Note 2). Nevertheless, we envision toolbox of synthetic organic chemistry to be useful for further diversifation of the products structures employing, for example, methyl-group in **2b**, F-atoms in **2d,e,h** and intrinsically reactive sites in **2i,j**. Common reactive Michael acceptors, such as methyl vinyl ketone and acrylonitrile, gave only trace amounts of CHCl$_2$ addition to the double bond as evidenced by GC-MS (Supplementary Note 3). Even more reactive Michael acceptors, i.e., methacrolein, methyl acrylate, and 2-furanone, gave a complex mixture and the desired products could not be identified (Supplementary Note 4).

Analysis of the substrates scope suggests that diarylsubstituted enones in general are more suitable substrates for photocatalytic dichloromethylation than terminal alkenes. Nevertheless, the advantages of the developed method are a simpler protocol and safer conditions. For example, the synthesis of **2a** and **2l** was

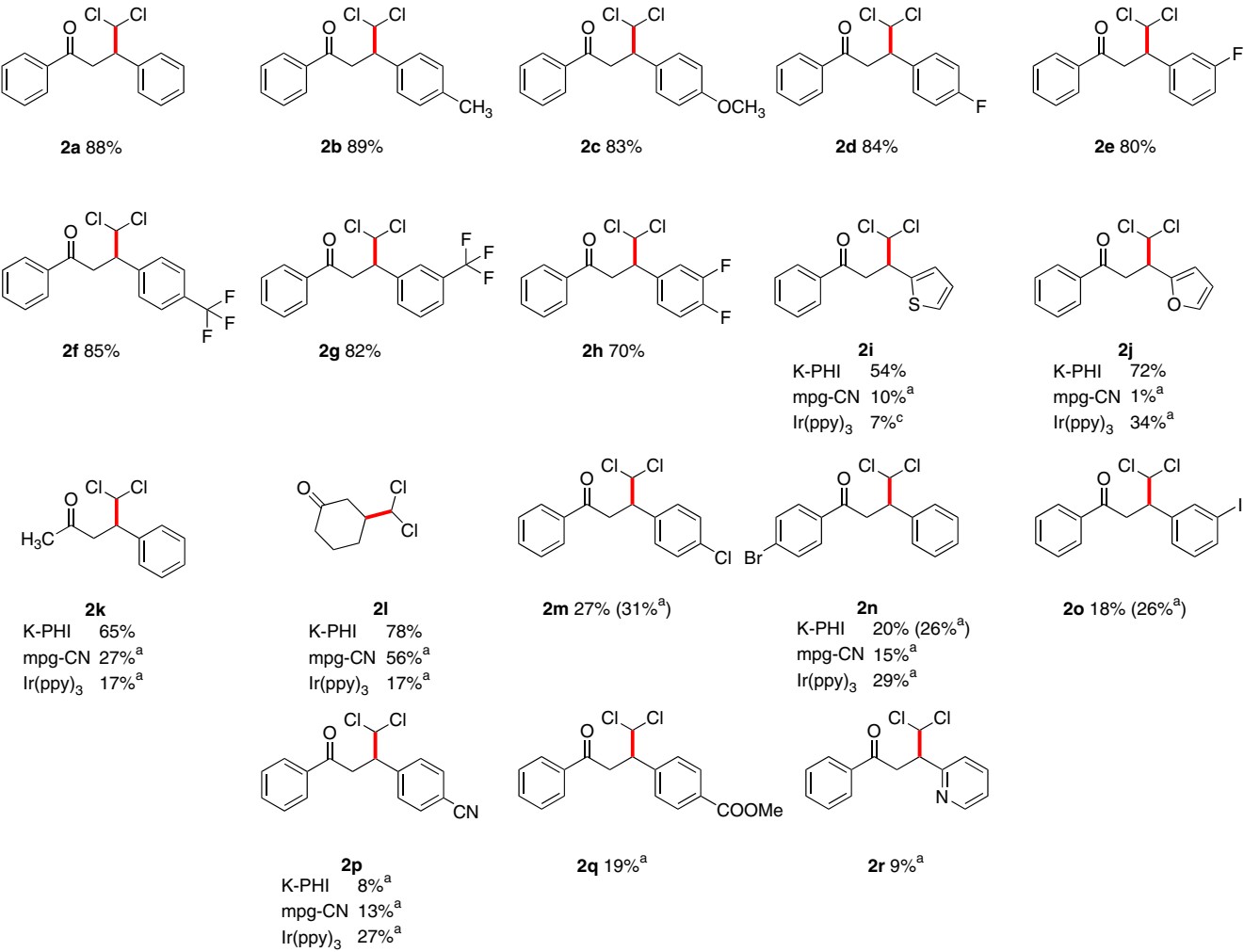

**Fig. 2 Scope of enone substrates.** Reaction conditions: enone 1 (1 eq., 0.05 mmol); TEOA (10 eq., 0.5 mmol, 67 μL); CHCl$_3$ (2 mL); Argon atmosphere; under light irradiation (λ = 461 nm, 51 ± 0.03 mW cm$^{-2}$, blue LED), isolated yields. Superscript "a" indicates the yields determined by $^1$H NMR using 1,2,3-trimethoxybenzene as internal standard.

described earlier using exotic reagents such as dichloromethyllithium[44]. A comparison of K-PHI, mpg-CN, and Ir(ppy)$_3$ photocatalysts using selected enones, **1i–l**, **n**, **p**, revealed that K-PHI in general gives the products in higher yields.

In the course of studying suitable reagents to install C$_x$Hal$_y$H$_z$-groups in the enone **1a**, we tested other halogenated reagents, including dichloromethane, bromoform, iodoform, 1,1,2,2-tetrachloroethane and tetrachloromethane (Supplementary Table 1). Careful analysis of the reaction mixture revealed that addition of CHBr$_2$-groups, CHI$_2$-groups, and C$_2$HCl$_4$-groups to enone **1a** indeed took place. However, the products are not stable and undergo further chemical transformations, such as HX elimination and subsequent cyclizations to 2,4-diphenylfuran (in case of bromoform and iodoform) or dichlorodihydropyranes (in case of tetrachloroethane) (Supplementary Note 5). Compared to bromoform and iodoform, chloroform is beneficial due to higher selectivity in the reaction of enones C1 backbone extension.

Scaling the dichloromethylation reaction of **1a** in batch led to gradual decrease of the dichloroketone yield, from 88% (on 0.05 mmol scale) to 23% (on 5 mmol scale) (Supplementary Table 3). After careful investigation, we concluded that the origin for such drastic drop of the product **2a** yield is poor light penetration in the depth of the batch reactor due to light scattering by suspended semiconductor particles (Supplementary Note 6)[45].

**Quasi-homogeneous flow photoreactor**. In order to overcome limitations of the batch approach, we performed the reaction in a continuous flow reactor made out of thin (inner diameter 1.6 mm) fluorinated ethylene propylene (FEP) tubing (Fig. 3). The use of carbon nitrides has been reported in serial micro-batch reactors[2], that use gas-liquid segments to avoid clogging. A simplified system is applicable for K-PHI due to relatively small particle diameter (100 nm) and negative zeta-potential (ζ) (Fig. 3a). Electrostatic stabilization allows pumping colloidal solution (Fig. 3b and Supplementary Note 7) without using a gas-liquid system (Fig. 3c). Using flow approach, **2a** was obtained with 57% yield. Considering convenience of K-PHI suspension pumping through thin FEP tubing along with easiness of the photocatalyst recovery, the entire system enables quasi-homogeneous photocatalysis in flow[39].

As seen from the light intensity measurements (Fig. 3d–f), the FEP tubing filled with the reaction mixture absorbs 74% [($I_0 - I_{T2})/I_{T0}$] of light. These results suggest that by performing the reaction in flow, more homogeneous irradiation of K-PHI particulate is achieved compared to the reaction in batch (Supplementary Note 6). Furthermore, we solved the problem of poor light permeability through a semiconductor suspension and increased the productivity of γ,γ-dichloroketone **2a** synthesis by a factor of 19.

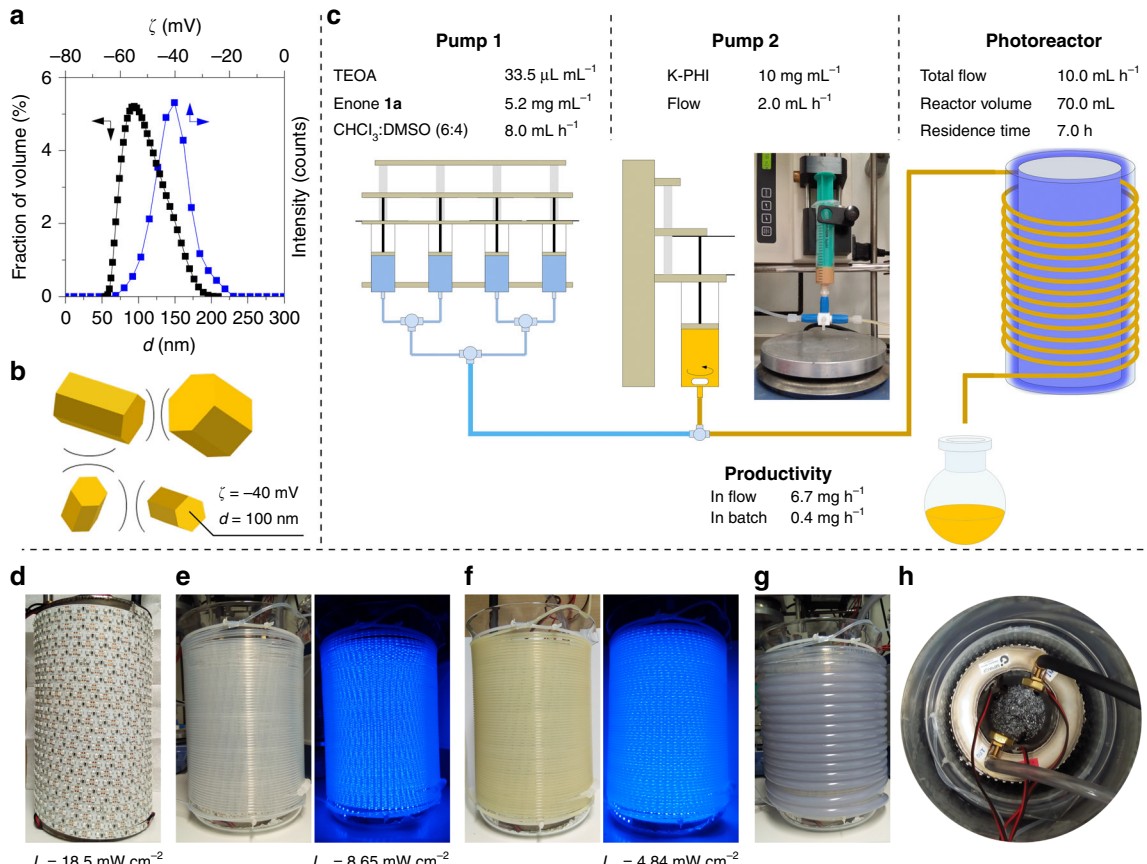

**Fig. 3 K-PHI colloidal solution properties and flow photoreactor design used in this work. a** Zeta-potential ($\zeta$) and hydrodynamic diameter ($d$) of K-PHI particles suspended in water. Source data are provided as a Source Data file. **b** Schematic representation of colloidal solution stabilization by electrostatic repulsion. **c** Schematic representation of the reactor setup and reaction parameters. Yield was determined by GC-MS collecting the solution for 30 min from the photoreactor; comparison between flow and batch are made considering the optimized reaction conditions in batch. **d** The light source was built by wrapping self-adhesive LED stripes around a hollow steel cylinder shell equipped with inlet and outlet for water-cooling. Incident light intensity ($I_O$) measured at zero distance from the light source 18.5 mW cm$^{-2}$. **e** Light source placed inside a glass beaker wrapped with FEP tubing under day light and blue light source. Transmitted light intensity ($I_{T1}$) was measured at zero distance from the FEP tubing. **f** FEP tubing filled with a reaction mixture under day light and blue light source. Transmitted light intensity ($I_{T2}$) was measured at zero distance from the FEP tubing. **g** Photoreactor wrapped with PVC tubing to maintain the desired temperature during the experiment. **h** View from the top on the assembled flow photoreactor immersed into a glass beaker. The space between the beakers is filled with cooling agent (water).

**Fig. 4 Preparation of different heterocyclic bioactive compounds from enones.** Isolated yields are shown.

**Application of γ,γ-dichloroketones in organic synthesis.** Finally, the γ,γ-dichloroketones obtained by the photocatalytic generation and addition of dichloromethyl radicals to enones were used to install other reactive functional groups. As a practical example, dichloroketone **2a** was converted to β-formyl ketone **3a** by simple hydrolysis with 60% yield (Fig. 4). The ketoaldehyde **3a** was then transformed into multi-substituted

heterocycles (**4a–6a**) using microwave assisted condensations with a series of nucleophiles. For instance, furan and pyrrole scaffolds have been used in synthesis of bioactive substances[46,47].

**Mechanism.** To support the role of chloroform as electron acceptor, we developed a method for oxidative coupling of

benzylamines (Fig. 5)[48]. As example, we synthesized four imines with 83–100% yield. These results offer an alternative route for such transformations using chloroform as a solvent and electron acceptor (Supplementary Fig. 3 for detailed mechanism of amines coupling).

The proposed mechanism of the reported photocatalytic reaction is shown in Fig. 6. In the first step, K-PHI is excited by blue photons giving excited state of the photocatalyst (K-PHI*). Removal of an electron from TEOA by K-PHI* (reductive quenching of the photocatalyst), leads to the formation of the long-lived radical anion K-PHI•−, that has the typical deep green color[27,29]. Chloroform is subsequently oxidized by a single electron transfer event, forming the chloroform radical anion that eliminates a chloride anion resulting in a dichloromethyl radical. Addition of the dichloromethyl radical to the β-carbon atom of the enone gives intermediate i−1. Abstraction of hydrogen from TEOA leads to the desired product 2. While TEOA acts as hole scavenger, chloroform simultaneously acts as solvent and electron acceptor to complete the photocatalytic cycle, as it was already proposed by Chen et al.[49] It is also possible to detect traces of different chlorinated compounds, that result from side radical reactions of the dichloromethyl radical, via GC-MS. By running experiments in the absence of the enone, we observed the formation of halogenated compounds including tetrachloroethane that is likely formed via a homocoupling of dichloromethyl radicals (Supplementary Table 4; Supplementary Figs. 4, 5, 6).

## Discussion

In this work, we extended the library of small organic radicals available for photocatalytic synthesis to dichloromethyl radicals than can be conveniently generated from chloroform. The method was validated in a 1,4 addition of dichloromethyl radicals to enones. The process is photocatalyzed by the heterogeneous, metal free carbon nitride photocatalyst K-PHI. Fifteen γ,γ-dichloroketones were isolated in moderate to excellent yield. The possibility to use other polyhalogenated compounds such as bromoform, iodoform and 1,1,2,2-tetrachloroethane has been demonstrated. Light scattering by carbon nitride particles has been identified as limiting factor for scaling these transformations. The results suggest that, in a typical photocatalytic experiment using 2.5 mg mL$^{-1}$ of semiconductor particles, the penetration depth of light is only 3 mm. In polar solvent, such as DMSO, nanoparticles of K-PHI give stable suspension that was pumped through a thin (1.6 mm internal diameter) tubing. γ,γ-dichloroketone 2a has been also synthesized using quasi-homogeneous photoreactor. The γ,γ-dichloroketones obtained in this work were proved to be useful building blocks with applications in the synthesis of bifunctional compounds that can be used for the preparation of heterocyclic bioactive molecules. The use of chloroform as solvent and electron acceptor was also demonstrated in the oxidative coupling of benzylamines.

## Methods

**Microwave reactions**. Experiments were carried out in a CEM Discover® SP System microwave reactor.

**Compounds characterization**. $^1$H and $^{13}$C NMR spectra were recorded on Agilent 400 MHz (at 400 MHz for Protons and 101 MHz for Carbon-13). Chemical shifts are reported in ppm versus solvent residual peak: chloroform-$d$ 7.26 ppm ($^1$H NMR), 77.1 ppm ($^{13}$C NMR); acetonitrile-$d_3$ 1.94 ppm ($^1$H NMR), 118.3 ppm ($^{13}$C NMR).

Mass spectral data were obtained using Agilent GC 6890 gas chromatograph, equipped with HP-5MS column (inner diameter = 0.25 mm, length = 30 m, and film = 0.25 μm), coupled with Agilent MSD 5975 mass spectrometer (electron ionization).

**Electrochemistry**. Cyclic voltammetry (CV) measurements were performed in a glass single-compartment electrochemical cell. Glassy carbon (diameter 3 mm) was used as a working electrode (WE), Ag wire in AgNO$_3$ (0.01 M) with tetra-butylammonium perchlorate (0.1 M) in MeCN as a reference electrode (RE), Pt wire as a counter electrode. Each compound was studied in a 50 mM concentration in a 0.1 M tetrabutylammonium perchlorate (TBAP)/chloroform electrolyte solution (10 mL). Before voltammograms were recorded, the solution was purged with Ar, and an Ar flow was kept in the headspace volume of the electrochemical cell during CV measurements. A potential scan rate of 0.050 V s$^{-1}$ was chosen, and the potential window ranging from +2.5 V to −2.5 V (and backwards) was investigated. Cyclic voltammetry was performed under room-temperature conditions (~20–22 °C).

**1′c**: R = H, X = CH **2′c**: R = H, X = CH 100%
**1′d**: R = Me, X = CH **2′d**: R = Me, X = CH 100%
**1′e**: R = OMe, X = CH **2′e**: R = OMe, X = CH 100%
**1′f**: R = H, X = N **2′f**: R = H, X = N 83%

**Fig. 5 Oxidative coupling of benzylamines.** GC-MS yields are shown.

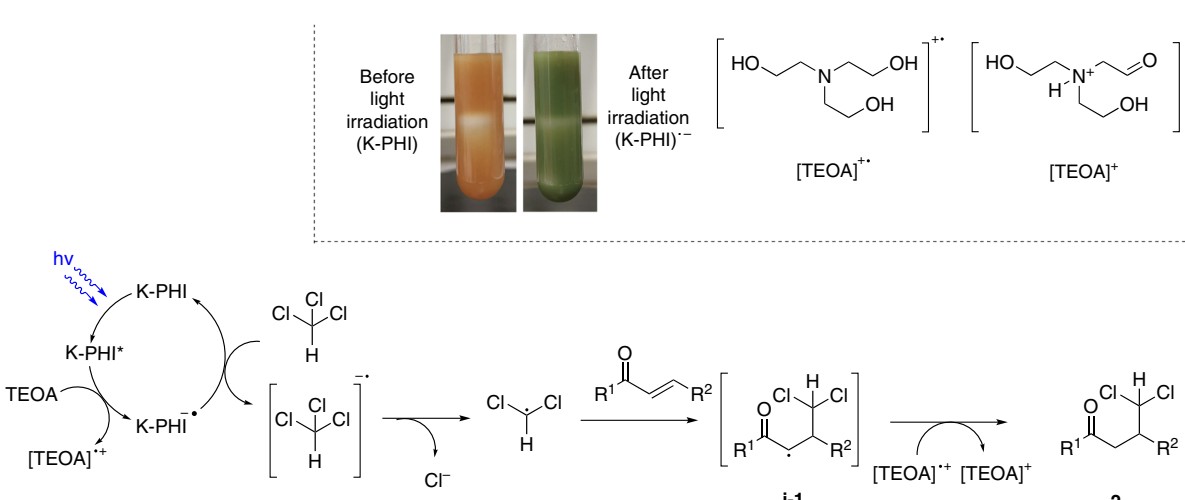

**Fig. 6 Proposed mechanism of the generation of dichloromethyl radicals and their addition to enones.** Inset shows images of the reaction mixture before and after light irradiation and structures of TEOA oxidation products.

**Photocatalysts characterization**. Zeta-potentials were measured in aqueous colloidal solution of K-PHI using a Malvern Zetasizer instrument.

Hydrodynamic diameter of K-PHI particles in water was measured using Malvern Zetasizer instrument.

**General method for dichloro-ketone preparation (2a–l)**. A glass tube with rubber-lined cap was evacuated and filled with argon three times. To this tube triethanolamine (74.6 mg, 66 μL, 0.5 mmol), corresponding chalcone (50 μmol), K-PHI (5 mg) and chloroform (2 mL) were added. Resulting mixture was stirred at 50 °C under irradiation of Blue LED ($\lambda = 461$ nm) for 20 h. Then reaction mixture was cooled to room temperature and centrifuged, clear solution was separated and solid residue was washed with chloroform (2 mL) and centrifuged again. Organic solutions were combined and evaporated to dryness. Residue after evaporation was purified by silica gel column chromatography using mixture of hexane/diethyl ether (98:2) as an eluent.

## Data availability

The data that support the findings of this study are available from the corresponding author upon reasonable request. The source data underlying Fig. 2a and Supplementary Fig. 1a–j are provided as a Source Data file.

## Code availability

This study does not use custom computer code or algorithm to generate results that are reported in the paper and central to its main claims.

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

## Acknowledgements
We gratefully acknowledge the Max-Planck Society for generous financial support. The Deutsche Forschungsgemeinschaft is gratefully acknowledged for providing financial support for this project (DFG-An 156 13-1). B.P. acknowledges financial support by a Liebig Fellowship of the German Chemical Industry Fund (Fonds der Chemischen Industrie, FCI). The authors thank Olaf Niemeyer (the head of NMR facility of the MPICI), Michael Born (assembly of LED), Marco Bott (fabrication of the steel cylinder for light source) for technical and scientific support.

## Author contributions
S.M. synthesis of precursors, photocatalytic tests, chemical properties of γ,γ-dichloroketones, flow photoreactor, preparation of manuscript and ESI; B.K. synthesis of precursors, photocatalytic tests; B.P. flow photoreactor; M.A. planning of research work; A.S. planning of research work, preparation of manuscript and ESI.

## Competing interests
The authors declare no competing interests.
