## [Peer Review File · Nature Communications]

Reviewers' comments:

Reviewer #1 (Remarks to the Author):

Carbon nitrides are an important class of heterogeneous photocatalysts for organic transformations because of their low cost synthesis, easy preparation and stability with reactive intermediates. Development of efficient synthetic methodologies using carbon nitrides as photocatalysts continues to attract considerable research effort from the synthetic chemists. Potassium poly(heptazine imide), a member of carbon nitrides family, can give metastable long-lived radical that has been used as a pool of electrons to reduce different substrates. In this manuscript, as a continuation of their previous reported photocatalytic methods using potassium poly(heptazine imide) as photocatalyst, Savateev and co-workers described an interesting 1,4-radical addition of dichloromethyl moiety to enones for formation of C-C bond using chloroform as a radical source. Under the optimal reaction conditions, the dichloromethylation products are obtained in 54-89% yields. The corresponding dichloromethylketones could be further transformed to multi-substituted heterocycles with a series of nucleophiles in two steps. A reasonable mechanism is also proposed for the reaction. Taken together, this chemistry represents a significant advance in the field of heterogeneous photochemistry, and should be attractive to the synthetic community.

Moreover, the manuscript is well written, and the supporting information is thorough and provides all the expected data. In summary, I find this work to be technically sound and interesting; however, it cannot meet the novelty, urgency and broad readership of highly impacted Nature Communication in its current form. However, this work might be suitable for acceptance to this journal if the authors can successfully address the follow points.

- 1) My major concern is about the substrate scope of the current methodology. Though a range of representative examples of enone substrates were demonstrated in Scheme 1, I do not see any other common Michael acceptors, like simple methyl vinyl ketone, acrylonitrile, acrylaldehyde, methyl acrylate, furan-2(5H)-one, and methacrylaldehyde, etc. Further application of the current method to these common substrates would significantly improve the functional group tolerance.
- 2) As for the aryl group in enones shown in Scheme 1, only those containing the methyl, methoxyl, fluoro and trifluoromethyl groups on the para- and meta-position are tested. Other common and useful functional groups like ester, nitrile, iodide, bromide, chloride on the aromatic ring and heterocyclic groups (pyridine, indole, etc) are not shown in the paper.
- 3) The yield of 2a in the gram-scale reaction is only 23%, much lower than 0.05 mmol scale (88%). It would be better if the yield could be improved to prove the robustness of the reaction. Or relative discussions need to be provided to explain this observation. In addition to chloroform, can other analogues be used as radical sources?

Reviewer #2 (Remarks to the Author):

The manuscript "Small Functional Organic Radicals: Dichloromethylation of Enones by Carbon Nitride Photocatalyst" by Mazzanti and coauthors report the photo-driven dichloromethylation of 12 enones using polymeric carbon nitride based photocatalyst. This is realized by employing chloroform and alcohol as the dichloromethyl radical source and hole scavenger. The authors have further employed the photogenerated dichloroketone to generate β -formyl ketone via hydrolysis, which was then transformed into multi-substituted heterocycles via condensation reaction with a series of nucleophiles. The paper is interesting in general, however from Entry 9-14 of Table 1 it seems that most commonly used heterogeneous photocatalysts (TiO₂, CdS, mpg-CN) and homogeneous photocatalyst worked reasonably well compared to that of K-PHI material. Therefore it is curious to me that why the authors only focus on K-PHI photocatalyst.

1. Stability test of the catalyst should be provided.
2. TEOA is used as hole scavenger and H source, why not use simple alcohols (e.g., ethanol, isopropanol)?
3. The photocatalytic reactions are performed under 50 oC. Will it still work under room temperature? Is it a thermo- or photo-catalytic process?
4. The quality of all figures and table are poor and difficult to read.

Replies to the referees' comments to the manuscript "Small Functional Organic Radicals: Dichloromethylation of Enones by Carbon Nitride Photocatalysis" (NCOMMS-19-27024-T) by Stefano Mazzanti, Bogdan Kurpil, Bartholomäus Pieber, Markus Antonietti and Aleksandr Savateev*

Reviewers' comments:

Reviewer #1 (Remarks to the Author):

Carbon nitrides are an important class of heterogenous photocatalysts for organic transformations because of their low cost synthesis, easy preparation and stability with reactive intermediates. Development of efficient synthetic methodologies using carbon nitrides as photocatalysts continues to attract considerable research effort from the synthetic chemists. Potassium poly(heptazine imide), a member of carbon nitrides family, can give metastable long-lived radical that has been used as a pool of electrons to reduce different substrates. In this manuscript, as a continuation of their previous reported photocatalytic methods using potassium poly(heptazine imide) as photocatalyst, Savateev and co-workers described an interesting 1,4-radical addition of dichloromethyl moiety to enones for formation of C-C bond using chloroform as a radical source. Under the optimal reaction conditions, the dichloromethylation products are obtained in 54-89% yields.

The corresponding dichloromethylketones could be further transformed to multi-substituted heterocycles with a series of nucleophiles in two steps. A reasonable mechanism is also proposed for the reaction. Taken together, this chemistry represents a significant advance in the field of heterogeneous photochemistry, and should be attractive to the synthetic community.

Moreover, the manuscript is well written, and the supporting information is thorough and provides all the expected data. In summary, I find this work to be technically sound and interesting; however, it cannot meet the novelty, urgency and broad readership of highly impacted Nature Communication in its current form. However, this work might be suitable for acceptance to this journal if the authors can successfully address the follow points.

Response: We appreciate referee's positive evaluation of our work. Below we provide point by point replies to the comments.

1) My major concern is about the substrate scope of the current methodology. Though a range of representative examples of enone substrates were demonstrated in Scheme 1, I do not see any other common Michael acceptors, like simple methyl vinyl ketone, acrylonitrile, acrylaldehyde, methyl acrylate, furan-2(5H)-one, and methacrylaldehyde, etc. Further

application of the current method to these common substrates would significantly improve the functional group tolerance.

Respond: Indeed, dichloromethyl functionalized Michael acceptors **2s-2w** are highly attractive building blocks for synthetic organic chemistry. As recommended by the referee we tested the following compounds as potential reagents in the reported dichloromethylation reaction under the conditions identical to that listed in the Scheme 1 of the manuscript:

The reaction mixture was analyzed by GC-MS. Please note that all Michael acceptors gave quite complex mixture. Therefore, we provide only mass spectra of the dichloromethylated products that we were able to identify. Assignment of the specific structure to the chromatogram peak was made taking into account isotope distribution of the molecular ion (denoted as Exp. or E. in the figures below). Theoretical isotope distribution (denoted as Theor. or T. in the figures below) was calculated using ChemDraw 17.1.0.105 (19) software.

In the GC-MS of the reaction mixture using methylvinyl ketone as a substrate, we identified 1) tetrachloroethane – the product of two dichloromethyl radicals coupling, 2) product of triethanol amine dehydrogenation,¹² 3) residual triethanolamine and 4) a small amount of dichloromethylated product **2s**. Isotope distribution in the mass spectrum of **2s** molecular ion matches well with the calculated one.

GC-MS chromatogram (a) of the reaction mixture using methyl vinyl ketone **1s** as a reagent in dichloromethylation reaction. Main reaction mixture components peaks are labeled. b) Mass-spectrum of **2s** radical cation with the theoretical m/z value. c) Zoomed in area of the mass spectrum. Experimental and theoretical m/z intensities with respect to the intensity of **2s** radical cation (m/z 154.0) are shown.

In the GC-MS of the reaction mixture using acrylonitrile as a substrate, we identified 1) tetrachloroethane – the product of two dichloromethyl radicals coupling, 2) product of triethanol amine dehydrogenation,¹² 3) residual triethanolamine and 4) a small amount of dichloromethylated product **2t**. Isotope distribution in the mass spectrum of **2t** molecular ion matches well with the calculated one.

GC-MS chromatogram (a) of the reaction mixture using acrylonitrile **1t** as a reagent in dichloromethylation reaction. Peaks in GC of the main components of the reaction mixture are labeled. b) Mass-spectrum of **2t** cation with the theoretical m/z value. c) Zoomed in area of the mass spectrum. Experimental and theoretical m/z intensities with respect to the intensity of **2t** cation (m/z 136.0) are shown.

Any attempts to separate compounds **2s** and **2t** from the reaction mixture by column chromatography were not successful (Supplementary Note 4).

Other Michael acceptors did not give the desired product even in trace quantity.

We assume that high reactivity of the Michael acceptors **1s-1w** is responsible for either low or no yield of the products **2s-2w**. Many of compounds **1s-1w** are used as polymer precursors due to reactive C=C bond. Therefore, in the photocatalytic reaction that involves radical intermediate, they most probably undergo polymerization.

Another reason for low or no yield of the products **2s-2w** may consist in high reactivity of these compounds toward nucleophiles, e.g. hydroxyl groups of triethanolamine during synthesis, silicagel hydroxyl groups during purification. Worth mentioning that reversed phase chromatography (C18 modified stationary phase, eluent – acetonitrile:water in different ratios) was not suitable for dichloroketones purification either.

Finally our conclusions about high reactivity of dichloromethylated compounds **2s-w** are supported by absence of almost any data regarding synthesis of these compounds obtained from SciFinder and Reaxys. To our surprise SciFinder output as of 24.10.2019 showed that only one article mentions methyl 4,4-dichlorobutanoate **2v** – a product of CHCl_2 addition to the β -carbon atom of methacrylate **1v**. [Acta Chemica Scandinavica, Series B: Organic Chemistry and Biochemistry, B35(3), 175-8; 1981] However, no NMR data is given for the title compound neither yield.

Compounds **2s-2u** and **2w** that we would have expected to obtain from the suggested Michael acceptors upon addition of CHCl_2 -moiety, have not been reported earlier.

SciFINDER CAS SOLUTION

Explore Saved Searches SciPlanner

Reaction Structure structure variable only at spe... > reactions (0)

REACTIIONS: REACTION STRUCTURE

Structure Editor: Java Non-Java

Search Type:

- Allow variability only as specified
- Substructure

Click image to change structure or view detail.

Import CIF

Search

Advanced Search

ChemDraw Launch a SciFinder substance or reaction

REFERENCES: Research Topic, Author Name, Company Name, Document Identifier, Journal, Patent, Tags

SUBSTANCES: Chemical Structure, Markush, Molecular Formula, Property, Substance Identifier

REACTIIONS: Reaction Structure

SciFINDER CAS SOLUTION

Explore Saved Searches SciPlanner

Explore Reactions resulted in 0 reactions Return Find Additional Reactions

Reaction Structure structure variable only at spe... > reactions (0)

REACTIIONS Find Additional Reactions

Analyze Refine

Analyze by: No reactions available

SciFINDER CAS SOLUTION

Explore Saved Searches SciPlanner

Reaction Structure structure variable only at spe... > reactions (0)

REACTIIONS: REACTION STRUCTURE

Structure Editor: Java Non-Java

Search Type:

- Allow variability only as specified
- Substructure

Click image to change structure or view detail.

Import CIF

Search

Advanced Search

ChemDraw Launch a SciFinder substance or reaction search directly!

REFERENCES: Research Topic, Author Name, Company Name, Document Identifier, Journal, Patent, Tags

SUBSTANCES: Chemical Structure, Markush, Molecular Formula, Property, Substance Identifier

REACTIIONS: Reaction Structure

SciFINDER CAS SOLUTION

Explore Saved Searches SciPlanner

Explore Reactions resulted in 0 reactions Return Find Additional Reactions

Reaction Structure structure variable only at spe... > reactions (0)

REACTIIONS Find Additional Reactions

Analyze Refine

Analyze by: No reactions available

SciFINDER CAS SOLUTION

Explore Saved Searches SciPlanner

Reaction Structure structure variable only at spe... > reactions (0)

REACTIIONS: REACTION STRUCTURE

Structure Editor: Java Non-Java

Search Type:

- Allow variability only as specified
- Substructure

Click image to change structure or view detail.

Import CIF

Search

Advanced Search

ChemDraw Launch a SciFinder substance or reaction

REFERENCES: Research Topic, Author Name, Company Name, Document Identifier, Journal, Patent, Tags

SUBSTANCES: Chemical Structure, Markush, Molecular Formula, Property, Substance Identifier

REACTIIONS: Reaction Structure

SciFINDER CAS SOLUTION

Explore Saved Searches SciPlanner

Explore Reactions resulted in 0 reactions Return Find Additional Reactions

Reaction Structure structure variable only at spe... > reactions (0)

REACTIIONS Find Additional Reactions

Analyze Refine

Analyze by: No reactions available

Reaction Structure substructure > reactions (17)

SciFinder

Explore Saved Searches SciPlanner

Reaction Structure substructure > reactions (17)

REACTIONS: REACTION STRUCTURE

Structure Editor:

Java Non-Java

Search Type:

- Allow variability only as specified
- Substructure

Click image to change structure or view detail.

Import CDF

Search

Advanced Search

SciFinder

Explore Saved Searches SciPlanner

Explore Reactions resulted in 0 reactions. Return Find Additional Reactions

Reaction Structure structure variable only at spe... > reactions (0)

REACTIONS

Analyze Refine

Analyze by:

No reactions available

Similar results were obtained from Reaxys.

Results for  New  Edit 
	0	Reactions	Reaction Query :  as drawn Edit in Query Builder Create Alert 	0	Reactions	Reaction Query :  average similarity; included: tautomers, only absolute stereo, additional ring closures allowed, salts, mixtures, isotopes, charges, radicals Edit in Query Builder Create Alert 
Structure x

Press ESC to close

Results for  New  Edit 
	0	Reactions	Reaction Query :  as drawn Edit in Query Builder Create Alert 	0	Reactions	Reaction Query :  average similarity; included: tautomers, only absolute stereo, additional ring closures allowed, salts, mixtures, isotopes, charges, radicals Edit in Query Builder Create Alert 
Structure x

Press ESC to close

Results for  New  Edit 
	0	Reactions	Reaction Query :  as drawn Edit in Query Builder Create Alert 	0	Reactions	Reaction Query :  average similarity; included: tautomers, only absolute stereo, additional ring closures allowed, salts, mixtures, isotopes, charges, radicals Edit in Query Builder Create Alert 
Structure X

Press ESC to close

Results for  New  Edit 
	0	Reactions	Reaction Query :  as drawn Edit in Query Builder Create Alert 	0	Reactions	Reaction Query :  average similarity; included: tautomers, only absolute stereo, additional ring closures allowed, salts, mixtures, isotopes, charges, radicals Edit in Query Builder Create Alert 
Structure X

Press ESC to close

Reaxys

Quick search Query builder Results Synthesis planner History

Stefano Mazzanti

Results for

New Edit

0 Reactions Reaction Query : as drawn
Edit in Query Builder Create Alert

0 Reactions Reaction Query : average similarity; included: tautomers, only absolute stereo, additional ring closures allowed, salts, mixtures, isotopes, charges, radicals
Edit in Query Builder Create Alert

Structure

Press ESC to close

ELSEVIER Copyright © 2019 Elsevier Ltd. All rights reserved. Terms and Conditions Privacy policy About content Performance Page Cookies are used by this site. To decline or learn more, visit our Cookies page RELX Group

Taking into account absence of any data regarding the dichloromethylated Michael acceptors **2s-2u** and **2w**, we conclude that synthesis of such compounds is not a trivial task. Despite high attractiveness for organic synthesis they have not been synthesized so far.

We need to point out that most of dichloroketones prepared in this work have not been reported either. However, success in dichloromethylation of enones bearing aromatic substituent we explain by higher stability of these molecules compared to small Michael acceptors under the reaction conditions. Due to steric hindrance and resonance stabilization of the aromatic substituents, diarylsubstituted enones are less susceptible for polymerization. Therefore, the path of CHCl_2 -moiety addition to the $\text{C}=\text{C}$ bond becomes dominant. Furthermore, isolation of dichloromethylated ketones is possible, but until the point when the structure becomes too electron deficient and hence susceptible for nucleophilic attack (see Supplementary Note 2).

These findings have been added to the 'Results and Discussion' section of the manuscript and ESI as a Supplementary Notes 3 and 4.

2) As for the aryl group in enones shown in Scheme 1, only those containing the methyl, methoxyl, fluoro and trifluoromethyl groups on the para- and meta-position are tested. Other common and useful functional groups like ester, nitrile, iodide, bromide, chloride on the aromatic ring and heterocyclic groups (pyridine, indole, etc) are not shown in the paper.

Respond: As suggested by the referee, we prepared a series of enones **1m-r** bearing other common and useful functional groups:

All these enones gave dichloroketones **2m-r** as evidenced by ^1H NMR spectra of the reaction mixture. Formation of these products was supported by the presence of doublet at 6.08-6.20 ppm in the ^1H NMR spectrum assigned to the CHCl_2 group, which is similar to the dichloroketone **2a-l**.

Our attempts to isolate CN-, CO_2Me - and pyridine-substituted dichloroketones **2p-r** by column chromatography were not successful.

Relatively low yields of dichloroketones **2p-r** bearing strong electron withdrawing groups (CO_2Me , CN, pyridine), we attribute to two factors. Firstly, electron withdrawing groups activate C=C bond for reduction, *i.e.* in the gas chromatograms of the reaction mixture we detected products of chalcones **1p-1r** C=C bond reduction. Similar behavior of electron deficient chalcones we observed earlier when studied their cyclodimerization.¹² In other words, addition of CHCl_2 -group to the C=C bond competes with the reduction of C=C bond.

Secondly, higher lability of dichloroketones **2p-r** toward nucleophiles creates difficulties for these compounds isolation by column chromatography due to irreversible interaction with OH-groups of silicagel. We also found that reversed phase chromatography (C18 modified stationary phase, eluent – acetonitrile:water in different ratios) is not suitable for γ,γ -dichloroketones purification either.

It should be pointed out that even a relatively weak nucleophile such as oxygen of the carbonyl group in dichloroketones **2** capable for triggering the intramolecular cyclization followed by elimination of two molecules HCl. Therefore, in the GC-MS (electron ionization) we have not detected a signal with m/z of dichloroketones, e.g. m/z 292 for **2a**, but we always observe signal of the furans generated from the corresponding dichloroketone, e.g. m/z 220 for 2,4-diphenylfuran.

Scheme 1 was updated with the structures of new products with isolated yields (**2m-o**) and NMR yields using internal standard (**2p-r**). ^1H and ^{13}C NMR spectra of dichloroketones **2m-o** after purification by column chromatography were added to the ESI. ‘Crude’ ^1H NMR spectra of dichloroketones **2p-r** have been added to the ESI as a Supplementary Note 2.

3) The yield of 2a in the gram-scale reaction is only 23%, much lower than 0.05 mmol scale (88%). It would be better if the yield could be improved to prove the robustness of the reaction. Or relative discussions need to be provided to explain this observation. In addition to chloroform, can other analogues be used as radical sources?

Respond: We performed a series of experiments in the batch reactor scaling up proportionally the amount of reagents and photocatalyst. The smallest amount of enone **1a** was 0.05 mmol and the largest amount – 5 mmol.

Table S3. Scale-up experiment.^a

Entry	1a, mmol (mg)	K-PHI, mg	TEOA, mL	CHCl ₃ , mL	Scale factor ^b	Time, h	Yield of 2a, ^c %
1	0.05 (10.4)	5	0.066	2	1	20	88
2	0.25 (52)	25	0.335	10	5	20	43
3	0.25 (52)	25	0.335	10	5	40	71
4	0.50 (104)	50	0.67	20	10	20	38
5	5.00 (1040)	250	6.6	100	100	20	23

^a Reaction conditions: blue LED ($\lambda = 461$ nm, 51 ± 0.03 mW cm⁻²); ^b with respect to enone **1a**; ^c determined by GC-MS

The yield of dichloroketone **2a** (η) decreases gradually as the amount (n) of reagents, i.e. chalcone **1a**, increases.

These results can be explained by the fact that the surface area of the liquid phase exposed to the light scales lower than the volume of the reaction mixture. Therefore, only a small fraction of the catalyst (close to the surface of the reaction) receives sufficient number of photons to mediate the reaction. The highest surface-area-to-volume ratio ($A = 4.57$) was for the reaction performed on 0.05 mmol scale, the lowest, $A = 1.37$, was for the reaction performed on 5 mmol scale. Assuming that the yield (η) of dichloroketone **2a** will approach zero in infinitely large photoreactor due to infinitely small number of photocatalyst located at the near surface layer and exposed to light, η versus A can be fitted with the linear function $\eta = 18.47 \times A$ ($R^2=0.995$).

In order to elucidate the origin of this effect we conducted further investigation.

Carbon nitride particles strongly absorb and scatter light.[Science, 2019, 365, 360-366, ESI, Figure S17-S18 in the ESI] Therefore, we concede that the chemical reaction occurs only in a thin layer located close to the surface of the liquid phase. On the other hand, the central area of the batch photoreactor remains in 'dark' and therefore no photocatalytic reaction occurs there. When reaction is performed on 0.05 mmol scale using 2 mL of solvent in a glass tube of 10 mm in diameter, most of the photoreactor volume is exposed to light, therefore average photoreactor productivity is high. On the other hand, on the 5 mmol scale using 200 mL of solvent in a glass tube of 30 mm in diameter, only a small volume of the batch photoreactor is effectively irradiated by light, while most of the reactor volume remains in dark. As a result average productivity of the photoreactor is low.

In order to find a distance from the reactor wall to the point in the bulk of the reaction mixture below which photons cannot penetrate, we performed light transmittance measurements of K-PHI suspension in CHCl_3 :DMSO (9:1) in 1 mm, 2 mm, 5 mm and 10 mm cuvettes using convenient UV-vis spectrometer. Ten percent of DMSO has been added to delay sedimentation of K-PHI particles during the measurements.

a) Transmittance (T) of the K-PHI suspension in CHCl_3 :DMSO (9:1) versus incident light wavelength measured using UV-vis spectrometer and LED emission spectrum (used in the photocatalytic experiments in this work); b) Transmittance (T) of the K-PHI suspension in CHCl_3 :DMSO (9:1) at $\lambda = 461$ nm versus cuvette optical path. Triangles denote data points obtained using UV-vis spectrometer. Squares denote data points obtained by measuring fraction of light ($\lambda = 461 \pm 20$ nm, $I_0 = 10.6$ mW cm^{-2}) from the external source passed through the cuvette filled with K-PHI suspension.

The data in the graphs above show that 1 mm thick layer of K-PHI suspension absorbs 94.50% of blue photons ($\lambda = 461 \text{ nm}$), 2 mm thick layer absorbs 99.15%, 5 mm thick layer absorbs 99.45% and 10 mm thick layer absorbs 99.50% of light. Transmittance measurements were corrected taking into account absorption of pure solvent.

Comparable results were obtained when we measured a fraction of light intensity ($461 \pm 20 \text{ nm}$, $I_0 = 10.6 \text{ mW cm}^{-2}$) passed through the cuvettes of variable length (1 mm, 2 mm, 5 mm and 10 mm) filled with K-PHI suspension. Figure below shows the experimental setup for measurement of light intensity transmitted through a 5 mm cuvette as an example.

a) 5 mm cuvette filled with CHCl_3 :DMSO (9:1) under day light; b) 5 mm cuvette filled with CHCl_3 :DMSO (9:1) under $461 \pm 20 \text{ nm}$ ($I_0 = 10.6 \text{ mW cm}^{-2}$) does not scatter light (no halo in the middle of the cuvette); c) Measurement of light intensity transmitted through a 5 mm cuvette filled with CHCl_3 :DMSO (9:1). Integrating sphere S142C connected to PM400 Optical Power and Energy Meter with readings on the left are shown; d) 5 mm cuvette filled with a suspension of K-PHI (2.5 mg mL^{-1}) in CHCl_3 :DMSO (9:1) under day light; e) 5 mm cuvette filled with a suspension of K-PHI (2.5 mg mL^{-1}) in CHCl_3 :DMSO (9:1) under $461 \pm 20 \text{ nm}$ ($I_0 = 10.6 \text{ mW cm}^{-2}$) scatters light strongly (strong halo in the middle of the cuvette); f) Measurement of light intensity transmitted through a 5 mm cuvette filled with a suspension of K-PHI (2.5 mg mL^{-1}) in CHCl_3 :DMSO (9:1). Integrating sphere S142C connected to PM400 Optical Power and Energy Meter with readings on the left are shown.

These reference measurements show that higher light intensity allows for better light delivery in the bulk of the reaction mixture. However, even using relatively high light intensity (10.6 mW cm^{-2}) compared to that in the UV-vis spectrometer, 95% of photons do not penetrate into the reaction mixture deeper than 3 mm.

Having this data, we calculated a specific absolute yield (N) of dichloroketone **2a** (mmol of dichloroketone **2a** produced by 1 gram of K-PHI) assuming that the 'active' volume of the batch

reactor is limited by 3 mm thick cylinder shell. Figure below shows specific absolute yield (N) of dichloroketone **2a** versus enone **1a** amount (n , mmol) taken for the experiment.

In this case the yield of **2a** is 7.05 ± 0.5 mmol g⁻¹ and depends weakly on the amount of enone **1a** taken for the experiment. Relative standard deviation in this case is 7.2%.

All in all, we conclude that the main limitation of scaling up the photocatalytic reaction mediated by the heterogeneous photocatalyst in batch is poor light penetration into the bulk of the reaction mixture.

In order to overcome limitations of the batch photocatalytic reactor, we performed a reaction in a continuous flow mode in a thin (inner diameter 1.6 mm) fluorinated ethylene propylene (FEP) tubing. In this case we replaced pure CHCl₃ by a mixture of CHCl₃:DMSO (3:2). It was done in order to keep K-PHI particles well dispersed in liquid medium and prevent FEP tubing from clogging. Image of the flow photoreactor we used in this project was added to the manuscript (Figure 2) and ESI. By using flow photoreactor, we obtained dichloroketone **2a** in 57% yield. Please note that comparable yield, *i.e.* 47% but on 0.05 mmol scale (Table S1, entry 23), was obtained when reaction was performed in batch using CHCl₃:DMSO mixture. Lower yield of the dichloroketone **2a** in CHCl₃:DMSO mixture compared to pure CHCl₃ we explain by possible interaction of the product with DMSO.

Several halogenated compounds have been investigated as possible sources of C_xHal_yH_z groups to introduce into enone **1a**. Reaction mixtures were analyzed by GC-MS. Assignment of the specific structure to the chromatogram peak was made taking into account isotope distribution of the molecular ion (denoted as Exp. or E. in the figures below). Theoretical isotope distribution (denoted as Theor. or T. in the figures below) was calculated using ChemDraw 17.1.0.105 (19).

The experiment using dichloromethane as solvent showed traces of the product. There are also intense peak of TEOA (broad due high concentration) and starting chalcone.

GC-MS data of the reaction mixture using dichloromethane as solvent.

In case of using bromoform, in the reaction mixture we identified the following major products: 1) residual bromoform, 2) compound with a brutto formula C_2HBr_3 , presumably 1,1,2-tribromoethene, 3) tetrabromoethene, 4) tetrabromoethane, 5) compound with a brutto formula C_9H_7OBr , presumably cinnamoyl bromide, 6) unreacted chalcone **1a**, 7) 2,4-diphenylfuran, 8) 4-bromo-1,3-diphenylbutan-1-one.

Formation of tetrabromoethane suggests that similarly to chloroform, in the photocatalytic reactor bromoform yields dibromomethyl radical. Two of such radicals recombine and give detectable products in GC-MS, *i.e.* tetrabromoethane, 1,1,2-tribromoethene.

Formation of 4-bromo-1,3-diphenylbutan-1-one suggests that addition of dibromomethyl radical to the C=C bond of enone **1a** took place. Reduction of the intermediary dihaloketone yields 4-bromo-1,3-diphenylbutan-1-one. Furthermore, intermediary dibromoketone undergoes intramolecular cyclization, *i.e.* nucleophilic attack of the carbonyl oxygen atom at the carbon atom of $CHBr_2$ -group, followed by elimination of two molecules HBr and yields 2,4-diphenylfuran.

This mechanism is further supported by the fact that no triethanolamine was detected in the reaction mixture by GC-MS – it has been converted to the salt. After the photocatalytic experiment we observed formation of gum-like residue. Therefore, only soluble fraction of the reaction mixture was analyzed by GC-MS.

GC-MS data of the reaction mixture using bromoform as solvent. a) Gas chromatogram. b) Mass spectrum of the GC peak corresponding to 4-bromo-1,3-diphenylbutan-1-one cation. c) Zoomed in mass spectrum of 4-bromo-1,3-diphenylbutan-1-one cation. d) Mass spectrum of the GC peak corresponding to 2,4-diphenylfuran radical cation. e) Zoomed in mass spectrum of 2,4-diphenylfuran radical cation. f) Mass spectrum of the GC peak corresponding to the compound with a tentative structure of cinnamoyl bromide radical cation. g) Zoomed in mass spectrum of cinnamoyl bromide radical cation.

In case of iodoform, i.e. solution in CH_2Cl_2 , we observed formation of small amount of 2,4-diphenylfuran implying that tentative CHI_2 -radical has been attached to the enone **1a**. Similarly to bromoform, no triethanolamine or its dehydrogenation products were detected in gas chromatogram suggesting acidification of the reaction mixture and subsequent binding triethanolamine to the insoluble salt.

GC-MS data of the reaction mixture using iodoform in dichloromethane. a) Gas chromatogram. b) Mass spectrum of the GC peak corresponding to 2,4-diphenylfuran radical cation. c) Zoomed in mass spectrum of 2,4-diphenylfuran radical cation.

Analysis of the GC-MS data of the reaction mixture using 1,1,2,2-tetrachloroethane as a solvent revealed that this halogenated solvent partially underwent chemical transformation.

GC-MS data of the reaction mixture using 1,1,2,2-tetrachloroethane as solvent. a) Gas chromatogram. b) Mass spectrum of the GC peak (retention time 10.950 min) corresponding to the compound with brutto formula $C_{17}H_{14}Cl_2O$ radical cation. One of possible chemical structures is shown. c) Zoomed in mass spectrum of $C_{17}H_{14}Cl_2O$ radical cation. d) Mass spectrum of the GC peak (retention time 10.996 min) corresponding to the compound with brutto formula $C_{17}H_{14}Cl_2O$ radical cation. One of possible chemical structures is shown. e) Zoomed in mass spectrum of $C_{17}H_{14}Cl_2O$ radical cation.

We identified the following products of tetrachloroethane conversion: 1) trichloroethene, 2) trichloroethane and 3) tetrachloroethene. *Cis*-isomer of chalcone **1a** was detected in the GC-MS of the reaction mixture. In addition, two compounds, presumably isomers, with m/z 304 and similar retention times, *i.e.* 10.650 min and 10.996 min, were detected. Based on the analysis of the isotope distribution in the molecular ion, we assigned brutto formula $C_{17}H_{14}Cl_2O$ to these compounds. The compounds with the brutto formula $C_{17}H_{14}Cl_2O$ might be the products of tetrachloroethane addition to the C=C bond of enone **1a** according to the scheme.

Elimination of two HCl molecules mediated by TEOA followed by one of the C=C bonds reduction yields the two products with brutto formula $\text{C}_{17}\text{H}_{14}\text{Cl}_2\text{O}$ that were detected in the GC-MS. Acidification of the reaction mixture was concluded based on the fact that TEOA was not detected by GC-MS. Instead it was converted to gum-like residue, similarly to the experiments with bromoform and iodoform, upon protonation with the generated HCl.

Reaction in tetrachloromethane led to isomerization of chalcone **1a**. Similarly to the experiments with bromoform, iodoform and tetrachloroethane, triethanolamine was converted to gum-like residue insoluble in tetrachloromethane.

GC-MS data of the reaction mixture using tetrachloromethane as solvent.

Overall, we conclude that K-PHI may be used to generate $\text{C}_x\text{Hal}_y\text{H}_z$ radicals from the respective halogenated compounds and potentially used to extend the backbone of enones.

The results of halogenated solvents screening experiments have been added to the SI as a Supplementary Note 5 and briefly discussed in the manuscript.

Reviewer #2 (Remarks to the Author):

The manuscript “Small Functional Organic Radicals: Dichloromethylation of Enones by Carbon Nitride Photocatalyst” by Mazzanti and coauthors report the photo-driven dichloromethylation of 12 enones using polymeric carbon nitride based photocatalyst. This is realized by employing chloroform and alcohol as the dichloromethyl radical source and hole scavenger. The authors have further employed the photogenerated dichloroketone to generate β -formyl ketone via hydrolysis, which was then transformed into multi-substituted heterocycles via condensation reaction with a series of nucleophiles. The paper is interesting in general, however from Entry 9-14 of Table 1 it seems that most commonly used heterogeneous photocatalysts (TiO₂, CdS, mpg-CN) and homogeneous photocatalyst worked reasonably well compared to that of K-PHI material. Therefore it is curious to me that why the authors only focus on K-PHI photocatalyst.

Response: We thank the referee for finding our paper interesting and positive evaluation of our work. Indeed, as stated in the Table 1 different heterogeneous photocatalyst and homogenous Ir(ppy)₃ successfully mediate the reported reaction. We focus on K-PHI, a type of carbon nitride material, primarily because it is cost-effective material. It is made of abundant and light elements, i. e. C, N, K. Therefore, in the long run prospective we envision no shortage in supply of elements needed to fabricate this photocatalyst. We attempt to create a more sustainable approach in photocatalysis. This is a research philosophy of the Innovative Heterogeneous Photocatalysis group (<http://www.mpikg.mpg.de/5785362/innovative-heterogeneous-photocatalysis>). Other advantages of using this material for photocatalysis we summarized in several reviews recently [ChemCatChem, 2019, doi: 10.1002/cctc.201901076; Eur. J. Org. Chem., 2019, 10.1002/ejoc.201901112]. References have been added to the introduction.

Below we give point-by-point replies to further referee’s comments.

1. Stability test of the catalyst should be provided.

Response: As suggested by the referee, we checked stability of K-PHI photocatalyst. Table S2 from the ESI suggests that after three cycles using the same photocatalyst the yield of dichloroketone **2a** remains at 97%. Relevant statement has been added to the manuscript.

2. TEOA is used as hole scavenger and H source, why not use simple alcohols (e.g., ethanol, isopropanol)?

Respond: In case of using ethanol as hole scavenger the yield of dichloroketone **2a** was 7 %, while *iso*-propanol gave the dichloroketone **2a** in 6 % yield. Such results may be explained by the fact that amines in general are better electron donors, i.e. in general they have lower oxidation potential, compared to alcohols. Thus, from cyclic voltammetry study we determined onset of triethanolamine oxidation to be +0.5 V vs Ag/AgNO₃, while for benzylalcohol +1.5 V vs Ag/AgNO₃ (Supplementary Note 1). For methanol, ethanol and *iso*-propanol we did not observe oxidation current when cyclic voltammetry measurements were performed in CHCl₃, suggesting that oxidation potential of these alcohols is more positive than oxidation of electrolyte, i.e. chloroform.

Cyclic voltammograms of different hole scavengers. From bottom to top: a) electrolyte $(n\text{Bu})_4\text{N}^+ \text{ClO}_4^-$ in chloroform (0.1M) purged with Ar; b) TEOA solution (50 mM) in electrolyte shows $E_{\text{ox}} = +0.5 \text{ V}$; c) BnOH solution (50 mM) in electrolyte shows $E_{\text{ox}} = +1.5 \text{ V}$.

From the experiments using alcohols as electron donors and data presented in Table 1, we infer that oxidation of triethanolamine occurs at nitrogen lone pair rather than hydroxyl group. In agreement with this hypothesis, triethylamine, for example, gave the dichloroketone **2a** in 37 % yield (Table S1, entry 9).

3. The photocatalytic reactions are performed under 50 °C. Will it still work under room temperature? Is it a thermo- or photo-catalytic process?

Respond: Indeed all dichloroketones reported in the manuscript were prepared at +50°C due to reactor heating when exposed to light from the LED. When reaction was performed at +20°C using forced cooling by ‘cold finger’ the yield of the dichloroketone **2a** was 64% (Table S1, entry 21). Therefore, we conclude that heating improves the yield of the dichloroketone. In order to simplify the experimental setup and save energy that otherwise would be spend to maintain lower temperature of the reaction mixture, we did not use forced cooling of the reaction mixture in most of cases.

4. The quality of all figures and table are poor and difficult to read.

Respond: We have improved the quality of all figures and table in the manuscript. We hope that now the manuscript looks better.

22 November 2019

Dr. Aleksandr Savateev

Reviewers' comments:

Reviewer #1 (Remarks to the Author):

I have reviewed an earlier version of this paper submitted by Savateev, Antonietti and co-workers which describes a 1,4-radical addition of dichloromethyl moiety to enones for formation of C-C bond using potassium poly(heptazine imide) as catalyst. Potassium poly(heptazine imide) as a member of carbon nitrides family can give metastable long-lived radical that has been used as a pool of electrons to reduce different substrates. And this manuscript is a continuation of their previous reported photocatalytic methods using potassium poly(heptazine imide) as photocatalyst. As previously mentioned, though this interesting methodology is in my eyes synthetically valuable, the substrate scope of the current methodology is still poor. The reaction of other common and useful functional groups like ester, nitrile, iodide, bromide, chloride on the aromatic ring and pyridine group gave low yields in the paper. On the other hand, besides chloroform, other halogenated reagents cannot be used as radical sources to realize 1,4-radical addition of enones.

For the above reasons, I consider that this manuscript is not suitable to be published in Nature Communication. I am of the opinion that any new method needs to show an advantageous niche of substrates in order to qualify for acceptance in a top journal.

Reviewer #2 (Remarks to the Author):

The revised manuscript has addressed all my comments to some extent, though I'm still not satisfied with the arguments regarding to my first and third comments. The use of K-PHI rather than gC₃N₄ or other photocatalysts seems nonsense to me, and oxidation of simple alcohols (ethanol and isopropanol) are as easy as TEOA.

Anyway, it sounds like an interesting work from a big group.

Reviewer #1 (Remarks to the Author):

I have reviewed an earlier version of this paper submitted by Savateev, Antonietti and co-workers which describes a 1,4-radical addition of dichloromethyl moiety to enones for formation of C-C bond using potassium poly(heptazine imide) as catalyst. Potassium poly(heptazine imide) as a member of carbon nitrides family can give metastable long-lived radical that has been used as a pool of electrons to reduce different substrates. And this manuscript is a continuation of their previous reported photocatalytic methods using potassium poly(heptazine imide) as photocatalyst. As previously mentioned, though this interesting methodology is my eyes synthetically valuable, the substrate scope of the current methodology is still poor. The reaction of other common and useful functional groups like ester, nitrile, iodide, bromide, chloride on the aromatic ring and pyridine group gave low yields in the paper. On the other hand, besides chloroform, other halogenated reagents cannot be used as radical sources to realize 1,4-radical addition of enones.

Reply: Supplementary Note 5 provides a detailed analysis of the reaction mixture using tetrachloromethane, iodoform, dichloromethane, tetrachloroethane and bromoform as radical sources in the reported reaction.

GC-MS of the reaction mixture using dichloromethane and iodoform shows that addition of the dihalomethyl radical to the C=C bond of enone **1a** proceeds, but much slower compared to the reaction with CHCl_3 . Bromoform gave a product of halogenated reagent addition to the C=C bond of the enone **1a**, however, selectivity of this reaction is lower compared to the reaction with CHCl_3 (Supplementary Note 5). Therefore, among halomethanes, chloroform is a more preferable source of dihalomethyl radicals in the enone backbone extension by one carbon atom.

Tetrachloroethane did give a product of radical addition to the C=C bond, presumably dichlorodihydropyrene (Supplementary Note 5).

In the first round of article revision, as requested by the review, we investigated reactivity of the enones with functional groups, Cl, Br, I, CN, CO_2Me , pyridine and isolated most of the products.

Our work reports 12 enones isolated with good to excellent yields 54-89%. In this work we also developed a setup to perform the reaction in flow. Therefore, γ,γ -dihaloketones **2b,d,e,i,j** are accessible in gram scale. The scope of products can be further diversified using available groups in the products **2b,d,e,i,j** and intrinsically reactive sites employing numerous methods of synthetic organic chemistry:

1. Methyl group in **2b** can be oxidized to alcohol using either chemical oxidants [Ashikari, Yoshida et al., *Org. Lett.* 2012, 14, 3, 938-941] or enzymes [Otomatsu et al. *Journal of Molecular Catalysis B: Enzymatic*, 2010, 66(1-2), 234-240]. Alternatively methyl group can be converted to aldehyde using chemical oxidants or photocatalytic methodology including various reports on carbon nitrides.[Chen, Wang et al., *Chem. Sci.* 2013, 4, 3244-3248; Kasap, Lotsch, Reisner et al, *J. Am. Chem. Soc.* 2016, 138, 9183-9192; Zhang, Mu, *Mater. Res. Bull.* 2014, 59, 84-92; Long, Wang, et al., *ChemSusChem* 2013, 6, 2074-2078; Chai, Xu et al., *J. Am. Chem. Soc.* 2016, 138, 32, 10128-10131] Alternatively methyl group in **2b** can be converted to thiomethyl group using also K-PHI photocatalyst.[Savateev et al., *Chem. Sci.* 2018, 9, 3584-3591]

2. Fluorine atoms either in the products **2d,e** or after conversion to the corresponding heterocycles as shown in Figure 5, can be employed in the reaction with nucleophiles [Lin, Wu, Cao et al., *Tetrahedron* 2017, 73, 1466-1472], [Knox, Chan et al., *ACS Chem. Biol.* 2018, 13, 1838–1843], [Li, Wu, Wang et al., *Bioorg. Med. Chem. Lett.* 2017, 27, 602–606]
3. Other functionalities can be easily introduced into compounds **2i** and **2j** bearing electron rich heterocycles, furan and thiophen, using one approach – electrophilic substitution reactions:
 - I-atom can be introduced by reaction with 1,3-diiodo-5,5-dimethylhydantoin [Iida, Arai et al., *J. Org. Chem.* 2019, 84, 7411–7417]
 - Br-atom can be introduced using N-bromosuccine imide.[Kawai et al., *Chem. Asian J.* 2014, 9, 2542 – 2547],[Mouri, Yamaguchi et al., *Chem. Sci.*, 2013, 4, 4465-4469], [Depaw, David-Cordonnier et al., *J. Med. Chem.* 2019, 62, 1306–1329]
 - Cl-atom atom using N-chlorosuccineimide.[Samanta, Yamomoto, *Chem. Eur. J.* 2015, 21, 11976–11979]
 - CN-group using BrCN.[Okamoto, Ohe et al., *Chem. Commun.*, 2012, 48, 3127-3129]
 - CO₂Me-group using lithiation followed by reaction with CO₂ and MeI.[Shigeno, Kondo et al., *Chem. Eur. J.* 2019, 25, 3235 –3239]

For the above reasons, I consider that this manuscript is not suitable to be published in Nature Communication. I am of the opinion that any new method needs to show an advantageous niche of substrates in order to qualify for acceptance in a top journal.

Reply: As declared by the editorial board of Nature Comminucations, the journal publishes not only articles related to synthetic organic chemistry, but ‘*high-quality research from all areas of the natural sciences*’. Novelty of our work consists not only in the developing of a new method, which in our opinion we have successfully accomplished (see reply to the comment above), but also includes the following aspects:

1. We use long-lived radicals of potassium poly(heptazine imide) generated by photocatalytic reduction of the semiconductor to trigger reduction of chloroform.
2. Our work shows advantage of using K-PHI not only due to recyclability and lower cost of the latter, the yields of γ,γ -dichloroketones are in general higher compared to mpg-CN and Ir(ppy)₃ (see also reply to the comment of reviewer #2).
3. K-PHI is represented by particles with diameter ~100 nm and negative zeta-potential that gives stable suspension in polar organic solvents. As such, morphology of K-PHI is employed to perform a reaction in quasi-homogeneous fashion – simply pumping a suspension of semiconductor through thin FEP tubing followed by separation using either centrifugation or filtration. 19 times higher productivity compared to the reaction in batch has been achieved.

Manuscript has been modified accordingly to emphasize on these central aspects of our work.

Reviewer #2 (Remarks to the Author):

The revised manuscript has addressed all my comments to some extent, though I'm still not satisfied with the arguments regarding to my first and third comments. The use of K-PHI rather than gC3N4 or other photocatalysts seems nonsense to me, and oxidation of simple alcohols (ethanol and isopropanol) are as easy as TEOA.

Anyway, it sounds like an interesting work from a big group.

Response: Below we provide point-by-point advantages of K-PHI compared to mpg-CN (a kind of gC3N4 with large surface area, in this work 200 m² g⁻¹) and Ir(ppy)₃ photocatalyst:

- 1. Higher yield of γ,γ -dichloroketones.** We performed a series of comparative tests using selected enones and heterogeneous mpg-CN and homogeneous Ir(ppy)₃ photocatalysts. Even though the yields of the parent dichloroketone **2a** are comparable for all photocatalysts (Table 1), enones **1i**, **1j**, **1k**, **1l** gave higher yields when K-PHI was used.

- 2. Stable colloidal solution.** Morphology of K-PHI is represented by particles with average diameter 100 nm. Together, with negative zeta-potential (−40 mV, *ACS Appl. Mater. Interfaces* **2017**, *9*, 22941-22949) K-PHI gives stable colloidal solution. Therefore, the reaction mixture can be conveniently pumped without danger of clogging the tubing. Photos below provide qualitative evidence that without agitation even after 6 h (longer than the contact time in the developed flow setup), large fraction of K-PHI particles remain suspended in CHCl₃:DMSO (3:2) mixture, while mpg-CN and g-C3N4 have almost completely precipitated after 1 h.

- 3. Recyclability.** Recyclability of the photocatalyst is becoming an important parameter.[Wen et al., *Green Chem.*, 2020, 22, 230-237; Xu et al., *Green Chem.*, 2020, 22, 136-143; Pieber et al., *Angew. Chem. Int. Ed.* 2018, 57 (31), 9976-9979] Our photocatalytic system remains active for at least 3 cycles.
- 4. Low price.** We estimated price of carbon nitride materials to be in the range 1-10 EUR g⁻¹ on gram scale synthesis,[Savateev, Antonietti, *ChemCatChem* 2019, 11, 6166-6176] while, for example, Ir(ppy)₃ costs 2204 EUR g⁻¹ (price from Sigma-Aldrich catalogue as of 17.01.2020, <https://www.sigmaaldrich.com/catalog/product/aldrich/694924?lang=de®ion=DE>). Therefore, choice of carbon nitrides, in particular K-PHI, for the developed reaction we also substantiate from the economical point of view.

Taking into account our own findings and literature data, we conclude that oxidation potential of TEOA is in fact lower compared to simple alcohols such as ethanol:

1. Cyclic voltammetry is considered by many research group as a primary technique to determine redox potentials of organic compounds.[Roth, Nicewicz et al., *Synlett* 2016, 27(05), 714-723; Patel, Molander et al., *ACS Catal.* 2017, 7, 3, 1766-1770] Our CV results clearly speak for much lower oxidation potential of TEOA ($E_{ox} = +0.5$ V vs Ag/AgNO₃) compared to benzylalcohol ($E_{ox} = +1.5$ V), ethanol, methanol and *iso*-propanol ($E_{ox} \sim 1.9$ V) in CHCl₃/ⁿBu₄N⁺ ClO₄⁻ electrolyte (Supplementary note 1).

- Higher rate of photocatalytic hydrogen evolution reaction have been reported for TEOA compared to alcohols as sacrificial electron donors. Thus, under the same conditions carbon nitride photocatalyst produces 15 times more H₂ using **TEOA compared to ethanol**. [Zhang et al. *Angew. Chem. Int. Ed.*, 2019, 58(42), 14950-14954; Figure S11 versus Figure S13 therein] In another report graphitic carbon nitride produces 14 times more H₂ using **TEOA compared to MeOH**. [Jones, Bowker et al., *Appl Catal. B*, 2019, 240, 373-379; Figure 3 therein] Given that amount of evolved H₂ is proportional to the amount of consumed electron donor, these results imply that TEOA is oxidized easier compared to alcohols. It is in agreement with textbook knowledge ‘... triethanolamine as an electron donor increases the hydrogen evolution rate by 3-5 times compared with methanol on account of its more negative redox potential.’ [Photocatalytic Hydrogen Production Using g-C₃N₄ in *Hydrogen Production Technologies* edited by Mehmet Sankir, Nurdan Demirci Sankir, John Wiley & Sons, 2017, pp. 656] All in all, the results of H₂ evolution rate using TEOA and alcohols as electron donors reported up to date agree with our findings – 97% yield of γ,γ -dichloroketone **2a** using TEOA, 10% for MeOH and 7% for EtOH (Table S1).
- Considering oxidation of benzylamine to imine and benzylalcohol to benzaldehyde as another indicative photocatalytic reaction, several papers report that under identical conditions (photocatalyst, time, light intensity, etc.) conversion of benzylic amines is higher compared to benzylic alcohols. [Bajada, Reisner et al., *ACS Appl. Mater. Interfaces* 2020, doi.org/10.1021/acsami.9b19718], [*ACS Sustainable Chem. Eng.* 2017, 5, 2562–2577]

The manuscript has been modified accordingly.

We thank the reviewers for their time and comments and you for your work as editor on our manuscript. We hope that with these changes our work is suitable for publication in Nature Communications.

Sincerely,

Aleksandr Savateev